# Assessing the Effect of a Major Quantitative Locus for Phosphorus Uptake (*Pup1*) in Rice (*O. sativa* L.) Grown under a Temperate Region

Ian Paul Navea [1], Jae-Hyuk Han [1,2], Na-Hyun Shin [1], O New Lee [3], Soon-Wook Kwon [4], Il-Ryong Choi [2] and Joong Hyoun Chin [1,*]

1   Department of Integrative Biological Sciences and Industry, Sejong University, 209 Neungdong-ro, Gwangjin-gu, Seoul 05006, Republic of Korea
2   The International Rice Research Institute—Korea Office, National Institute of Crop Science, Rural Development Administration, Iseo-myeon, Wanju-gun 55365, Republic of Korea
3   Department of Bioindustry and Bioresource Engineering, Plant Engineering Research Institute, Sejong University, Seoul 05006, Republic of Korea
4   Department of Plant Bioscience, College of Natural Resources and Life Science, Pusan National University, Milyang 50463, Republic of Korea
*   Correspondence: jhchin@sejong.ac.kr

**Abstract:** Water and phosphorus (P) fertilizer are two of the most critical inputs in rice cultivation. Irrigation and chemical fertilizers are becoming limiting factors under climate change and urbanization, which is leading to significant losses in yield. The *Pup1* quantitative trait locus (QTL) confers tolerance to P starvation through enhanced early-stage root vigor and P uptake in *indica* rice grown in the tropics. Whether the QTL works in temperate rice genetic backgrounds grown in temperate regions remains to be elucidated. To address this question, we introgressed the *Pup1* QTL into three temperate rice varieties—MS11, TR22183, and Dasanbyeo—using marker-assisted backcrossing and high-density genotyping. The selected lines all harbored the full *Pup1* QTL with recurrent parent genome recovery rates ranging from 66.5% to 99.8%. Under the rainfed and P non-supplied conditions, *Pup1* introgression lines did not show clear advantages over the recurrent parents in terms of vegetative growth and grain yield per plant, but exhibited enhanced yield responses to P application, except in Dasanbyeo, which a temperate rice that is genetically similar to *indica*. Our results suggest that *Pup1* confers enhanced P uptake in temperate rice and that the efficacy of *Pup1* might depend on the subspecific genomic background of the rice, whether it is *japonica* or *indica*.

**Keywords:** climate change; *Pup1*; marker-assisted breeding; abiotic stress; phosphorus uptake; drought

## 1. Introduction

To cope up with the increasing global rice demand due to population growth, cereal production must double in the next 25 years [1]. The impact of rice varieties that are well-adapted to low agricultural inputs is potentially big, as production constraints, such as the increase in seasonal temperature and drought resulting from climate change and increasing fertilizer cost, aggravate the economic challenges faced by rice farmers. Furthermore, urbanization has appropriated millions of hectares of crop land, and this trend is expected to persist as cities continue to expand [2]. As a result, there is a general tendency for farmers to cultivate rice in areas with limited access to proper irrigation and soil nutrition. In fact, about half of the rice cultivation regions in Asia are situated in problem soils and areas where irrigation is scarce [3]. Agriculture is the biggest consumer of water worldwide, accounting for 70% of fresh water demands. As the population increases, so does the demand for irrigation water. The increased demand could potentially compete with the demand from industrial and domestic sectors [4]. In addition, approximately 1.2 billion

people live and produce food in areas without proper irrigation, and this trend is expected to continue with population booms [5].

Phosphorus (P) is the second most important inorganic nutrient in plants after nitrogen. Approximately 5.7 billion hectares of land lack enough plant-available P, owing to P's immobility in soil [6]. Plants grown in P deficient soil appear smaller with dark green leaves, have retarded root growth, and have less tillers, which leads to a substantial drop in yield [7]. Farmers often try to correct P deficiency by applying additional fertilizers. However, this is not feasible in low input cultivation where farmers have insufficient resources and, in some occasions, do not have access to P fertilizers at all [8]. This problem worsens as fertilizer costs inevitably spike, since phosphorus fertilizers have finite natural sources, thus making the access to adequate fertilization more limited [9]. Furthermore, the excessive application of P fertilizer contaminates the ground water and leads to algal blooms and eutrophication [10].

Plants have evolved to adapt to P starvation. These mechanisms involve root interception of inorganic P (Pi), P uptake efficiency, and internal P use [11]. In a study using a diverse set of rice varieties, genetic variation on root shape and growth were detected, whereas no variation was observed on P uptake efficiency and internal P use. This implies that the improvement of root architecture is a potential target for improvement through molecular marker and genomics-assisted breeding [12].

*Pup1*, a major QTL conferring tolerance to P deficiency in rice, was identified in the long arm of chromosome 12 in two independent studies despite the difference in phenotyping methods [13,14]. *OsPSTOL1*, one of the genes harbored within the *Pup1* locus, encodes a protein kinase that enhances phosphorus uptake under P starvation through enhanced early-stage root vigor and crown root number in seedlings. This leads to improved yield in transgenic rice grown in P deficient soils [15]. Investigation on *Pup1*'s natural variation revealed that the QTL is present in the majority of upland varieties, whereas the opposite is true in the developed, low-land varieties [16]. Lowland-adapted, tropical *indica* introgression lines have been developed using a core set of molecular markers tagging the important genes within the *Pup1* QTL [16,17].

*Pup1 QTL* confers tolerance to P starvation in *indica* and *japonica* introgression lines grown under tropical conditions. In a contrasting set of Nipponbare near isogenic lines, *Pup1* increased P uptake by 3–4 folds relative to the RP, and resulted to a 2- to 4-fold higher grain yield per plant compared to *Pup1*-negative sister lines in P deficient soil [12]. In the same study, *Pup1* introgression lines of the IR64, IR74, Situ Bagendit, Batur, and Dodokan tropical varieties showed better yield when compared to their *Pup1*-negative sister lines. Based on the studies mentioned, *Pup1* works well when introgressed in rice grown in tropical conditions. However, knowledge is lacking on whether *Pup1* QTL works in a temperate rice genetic background grown in temperate regions. In addition, to date, Nipponbare is the only temperate *japonica* variety utilized in *Pup1* studies [13]. Therefore, there is a need to test *Pup1* in various *japonica* rice in temperate regions. To assess *Pup1*'s functionality in temperate conditions, we introgressed *Pup1* into three temperate varieties—MS11, TR22183, and Dasanbyeo—through molecular marker-assisted backcrossing and next-generation sequencing. The introgression lines were evaluated for important agronomic traits under low inputs of P and rainfed conditions in a temperate region.

## 2. Materials and Methods

### 2.1. Plant Materials

IR64-Pup1 (SMTA no. refer to [13]) was imported from the International Rice Research Institute (IRRI) and was utilized as the *Pup1* QTL donor (DP) to develop three temperate *Pup1* introgression rice breeding lines. The recurrent parents (RPs) included MS11, TR22813, and Dasanbyeo. MS11 is a tropical climate-adapted temperate *japonica* derived from a cross between Jinmibyeo and Cheolwon 46, also known as IRRI 142 and NSIC Rc 170 in the Philippines, and Asemi in South Korea. MS11 is lodging-resistant, with high grain quality and yield, and moderate resistance against rice blast and bacterial blight. However, it is

intolerant to P deficiency when grown in the tropics [14]. Dasanbyeo, a fertilizer input-responsive variety, is a Tongil-type rice, derived from intercrosses among IR8, Taichung Native 1, and Yukara (IR667-98-1-2). In terms of genomic structure, it is closely related to *indica*-type varieties [15]. TR22183, on the other hand, originated from Northeastern China and is clustered as temperate *japonica* using subspecies-specific STS markers [16]. TR22183 is a nitrogen-use efficient, cold-tolerant rice variety [17,18].

### 2.2. Pup1 Introgression into Temperate Rice through Marker-Assisted Backcrossing

*Pup1* QTL donor IR64-*Pup1* was crossed with MS11, TR22183, and Dasanbyeo, while utilizing the RPs as the female plant to recover both cytoplasmic and nuclear genomes. Molecular marker-assisted backcrossing (MABC) was employed in this breeding program (Figure 1).

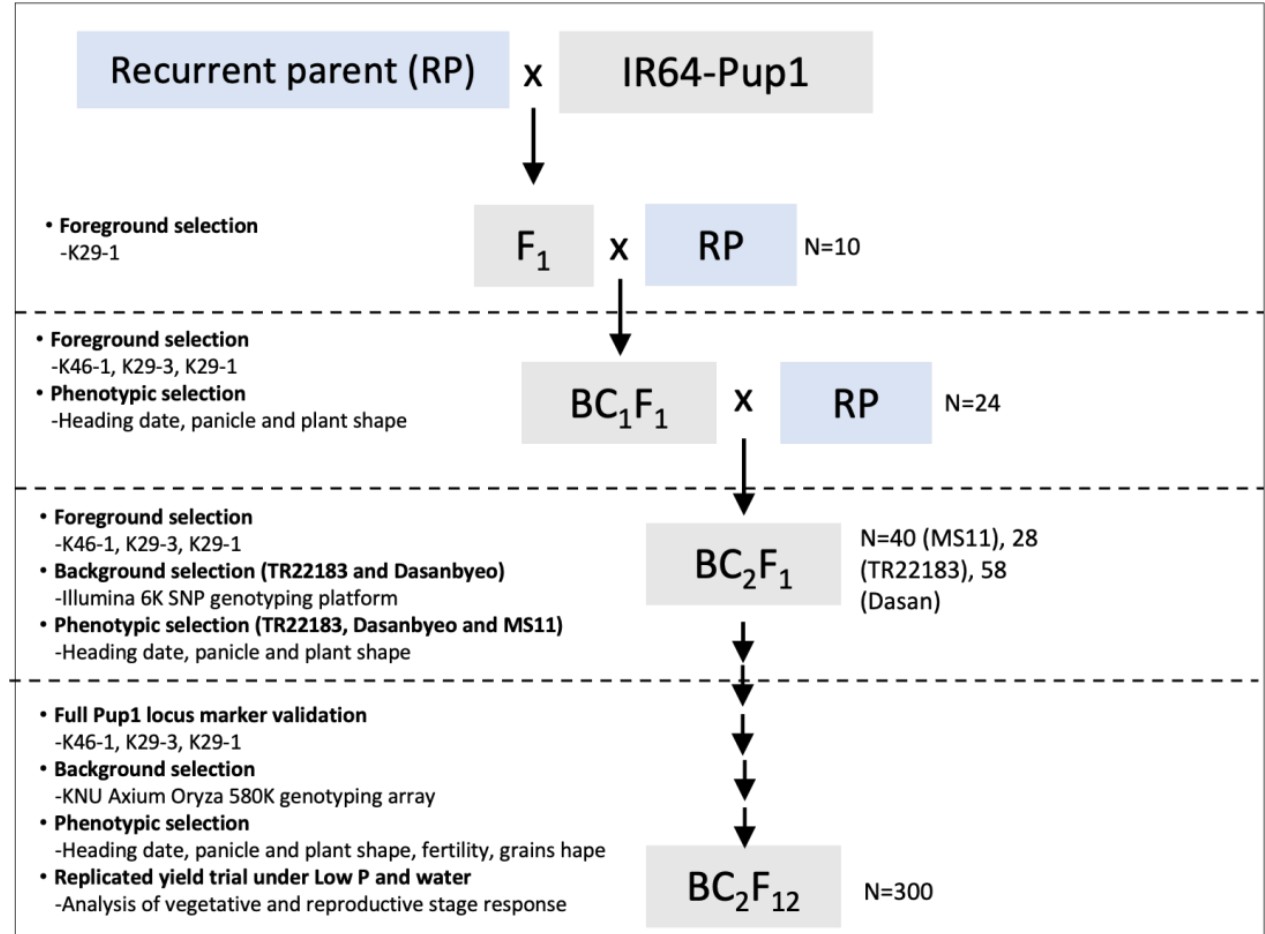

**Figure 1.** Marker-assisted breeding scheme for the introgression of *Pup1* into MS11, TR22183, and Dasanbyeo.

Three core markers tagging the three most important genes in the *Pup1* locus were utilized as foreground markers [19]. The marker set consisted of K46-1 (*OsPSTOL1*), K29-3 (*OsPupK20* and *OsPupK29-1*), and K29-1 (*OsPupK29-1*). Ten $F_1$ plants from each cross were subjected to foreground genotyping using the K29-1 codominant *Pup1* marker. Two successful $F_1$ plants from each combination were crossed with the RP to obtain $BC_1F_1$ plants. Twenty-four $BC_1F_1$ plants from each cross were foreground-genotyped using the core markers. Plants having *Pup1* donor alleles using the core markers were considered full *Pup1* introgression lines. Twelve, eleven, and ten $BC_1F_1$ plants harboring the full *Pup1* QTL were selected from MS11, TR22183, Dasanbyeo–IR64-*Pup1* combinations, respectively. The selected $BC_1F_1$ plants, together with the RPs, were grown in well-watered pots with

normal fertilizer application. At the pre-flowering stage, a single $BC_1F_1$ plant resembling the plant shape and heading date of the RPs was backcrossed to produce $BC_2F_1$ seeds.

Forty $BC_2F_1$ MS11 background *Pup1* introgression plants were subjected to foreground genotyping. The selected 10 plants with full *Pup1* introgression were grown in pots in well-watered and normal fertilizer conditions. A single $BC_2F_1$ plant containing full *Pup1* introgression resembling the parents in terms of plant shape, heading date, panicle, and grain shape was selected and advanced to $BC_2F_2$. Forty $BC_2F_2$ MS11 *Pup1* introgression lines were genotypically screened for a full *Pup1* introgression using foreground markers. Seven lines harboring the full *Pup1* QTL were selected and grown in pots, together with the RPs. Using the pedigree method, 2 plants were selected based on phenotypic similarity with MS11 and advanced to $BC_2F_{12}$.

In case of TR22183 and Dasanbyeo, a total of 28 and 58 $BC_2F_1$ plants were subjected to foreground genotyping, respectively, wherein a total of 15 (TR22183) and 27 (Dasanbyeo) plants with full *Pup1* introgression were selected for high-density background genotyping. A single plant harboring the full *Pup1* region with the highest RP genome recovery rate was selected and advanced to $BC_2F_{12}$ using the pedigree method from each cross combination. At $BC_2F_{12}$, five to ten plants from each introgression lines were further confirmed to have the presence of the homozygous full *Pup1* locus by using foreground markers, as well as for the RP genomic recovery rates.

### 2.3. Foreground and Background Genotypic Selection

Total genomic DNA (gDNA) was extracted from the leaf tissues of one-month-old seedlings using the Cetyl Trimethyl Ammonium Bromide (CTAB) method [20]. Genomic DNA was standardized to 50 ng/μL using a NanoDrop™ 2000/2000c Spectrophotometer (Thermo Fisher, Altrincham, UK). Polymerase chain reaction (PCR) in the foreground genotyping step was conducted using a PCR thermal cycler (SimpliAmp, Thermo Scientific, Altrincham, UK). The 20 μL PCR mixture consisted of 2 μL of 50 ng of gDNA, 0.5 μL of 10 pmol each of forward and reverse primers, 0.5 μL of 2.5 mM dNTPs, 2 μL of 10X reaction buffer, and 0.1 μL of 500 units/ μL of taq polymerase (Bioneer, Daejeon, Republic of Korea). Thermal cycling was conducted using the following profile: initial denaturation at 94 °C for 4 min, followed by 30 cycles of denaturation at 94 °C for 1 min, annealing at 55–68 °C for 30–60 s, initial extension at 72 °C for 60 s, and final extension at 72 °C for 7 min. PCR products were separated using agarose gel electrophoresis (BioFACT, Daejon, Republic of Korea) on 1.6–4% agarose gels at 120 V for 1–1.5 hrs. in 0.5X TBE buffer. The gels were stained with SYBR Safe (Invitrogen, Waltham, MA, USA). PCR amplicons were visualized in gels using a gel documentation system (Korea Lab Tech, Seongnam, Republic of Korea). Background genotyping was conducted using the Illumina 6K single nucleotide polymorphism (SNP) genotyping [21] platform (for TR22183 and Dasanbyeo introgression lines) and the KNU Axium Oryza 580K SNP Genotyping array (for MS11, TR22183, and Dasanbyeo) [22]. Polymorphic markers between RPs and the DP were used as the basis for the estimation of the background genome recovery rates. Percent background genome recovery was calculated as the number of markers with the RP allele divided by the total number of polymorphic markers, multiplied by one-hundred.

### 2.4. Field Trials and Experimental Design

The $BC_2F_{12}$ introgression lines and RPs were sown in plastic trays inside the greenhouse on the 20 May 2021 and on the 25 May 2022. Twenty-one days after sowing, the seedlings were transplanted into the experimental fields with a planting density of $15 \times 30$ cm at one seedling per hill. In 2021, field trials were conducted under irrigated (Gokseong, Jollanam-do, Republic of Korea, 35.30 North (N), 127.30 East (E)) and rain-fed (Greenhouse, Hwaseong City, Gyeonggi-do, Republic of Korea, 37.16 N, 126.82 E) conditions with normal fertilizer application (N-$P_2O_5$-$K_2O$ = 21-11-11 kg/10 a). In 2022, the plants were cultivated in a rice field in Korea (Hwaseong City, Gyeonggi-do, Republic of Korea, 37.16 N, 126.82 E) where the field was divided equally into two plots;

one applied with normal fertilizer rate while the other was treated without P fertilizer ($N-P_2O_5-K_2O$ = 21-0-11 kg/10 a). In 2021, a four to five cm water depth was maintained in the irrigated plots from transplanting until most of the rice were on ripening stage. In both years, water was withdrawn from the rainfed plots 30 days after transplanting (DAT). The irrigation was applied when visible cracks were observed in the soil. The experiments were laid out in a randomized complete block design with two replications, with each line consisting of three rows with 24 hills in each plot and replication. Composite soil samples from at least five random plot sites were collected and analyzed for physical and chemical properties (Cheil Research Center, Seoul, Republic of Korea) before transplanting, at the maximum tillering and maturity stage. All the lines were evaluated in both years, except for Dasanbyeo, which was not tested in 2021.

*2.5. Phenotyping and Identification of Low P and Water Tolerance Indices*

Five plants from the middle row of each line and RPs were selected for data collection. Vegetative data, including plant height (*ph*, cm), length from the ground to the longest leaf, tiller number (*tn*), and SPAD value were collected 30, 44, and 54 days after transplanting (DAT). Maturity stage data, including culm length (*cl*, cm); number of panicles (*pn*); panicle length (*pl*, cm); length of the 3 longest panicles per plant; one-hundred grain weight (*hgw*, g); weight of 100 filled grains; fertility (*fer*, %) from the three longest panicles with grains having approximately 14% moisture content; grain yield per plant (*gypp*, g); days to heading (*dth*, DAS); number of days after sowing when 50% of the total plants were heading were recorded and analyzed.

*2.6. Statistical Analysis of Phenotypic Data*

The data set was subjected to analysis of variance (ANOVA) using the general linear model. The effect of genotype, P application levels, and water regimes were evaluated. Traits were evaluated for responsiveness to different P–water applications. Pairwise mean comparison was conducted using the Duncan Multiple Range Test (DMRT) at $p = 0.05$. Statistical analyses were carried out using the Statistical Tool for Agricultural Research (STAR, IRRI) v2.0.1 which was developed based on Eclipse Rich Client Platform (RCP) and R Language v1.5.

**3. Results**

*3.1. Introgression of the Full Pup1 Locus and RP Genome Recovery Rates in Temperate Rice*

Plants were genotypically screened for *Pup1* introgression using three gel-based foreground core markers [23]. The selected plants all contained the target locus and harbored the tolerant (Kasalath) allele at the *Pup1* locus. $BC_2F_1$ plants in TR22183 and Dasanbyeo backgrounds were background genotyped using the 6K Illumina genotyping platform. Of the total 5,274 SNP markers, 2,055 were polymorphic between the TR22183 and IR64-Pup1 (*Pup1* donor), whereas 146 markers were polymorphic between the Dasanbyeo and IR64-Pup1. The highest RP genome recovery rates in the selected $BC_2F_1$ plants were 81.9% and 79.8% in the TR22183- and Dasanbyeo- Pup1 introgression lines, respectively (Table S3). The plant with the highest RP genome recovery rate was advanced to $BC_2F_{12}$ using the pedigree method.

In case of the MS11-Pup1, background genotyping was not conducted at the $BC_2F_1$. Instead, pedigree selection was done based on the morphological similarity to the RP. Two promising lines, in terms of phenotypic similarity to MS11, were advanced to $BC_2F_{12}$, namely the MS11-Pup1A and MS11-Pup1B.

At the pre-yield trial stage ($BC_2F_{12}$), the introgression lines were further confirmed for homozygosity at the Pup1 locus using the foreground core markers. The selected lines all harbored the homozygous full Pup1 donor allele (Figure 2). To estimate the RP genome recovery rate before yield trial, the selected $BC_2F_{12}$ lines were subjected to background genotyping using the KNU Axium 580K SNP platform (with reference to IRGSP v1.0). Of the total 584,000 SNP markers, 208,000 were polymorphic between MS11 and the DP. For

TR22183 and Dasanbyeo, 181,124 and 149,078 SNP markers were respectively polymorphic to the DP (Table S3). TR22183-Pup1 and Dasanbyeo-Pup1 had RP genome recovery rates of 99.75% and 92.33%, respectively. MS11-Pup1A and MS11-Pup1B had RP genome recovery rates of 71.68% and 66.50%, respectively. (Table 1 and Figure 3).

*3.2. Vegetative Stage Response of Temperate Pup1 Introgression Lines to Low Inputs of P and Water*

Introgression lines grown under low P and water input exhibited various vegetative stage response patterns depending on the genetic background (Table 2). MS11 plants were at least as tall as the two *Pup1* introgression lines, MS11-Pup1A and MS11-Pup1B, at 30 to 44 DAT. At 54 DAT, the MS11-Pup1B's plant height (*ph*) was greater than that of the MS11, whereas the MS11-Pup1A had the same *ph* as the MS11. The MS11-Pup1B consistently had more tillers and higher SPAD values compared to the MS11, except at 30 DAT. The MS11-Pup1A, however, did not show significant difference to the MS11 with regards to the number of tillers (*tn*) and SPAD readings throughout the early vegetative stage. In P-supplied soil, there was no significant difference in terms of *ph*, although the MS11-Pup1B (11.33) had significantly more tillers than the MS11 (7.50) towards the end of the vegetative growth stage.

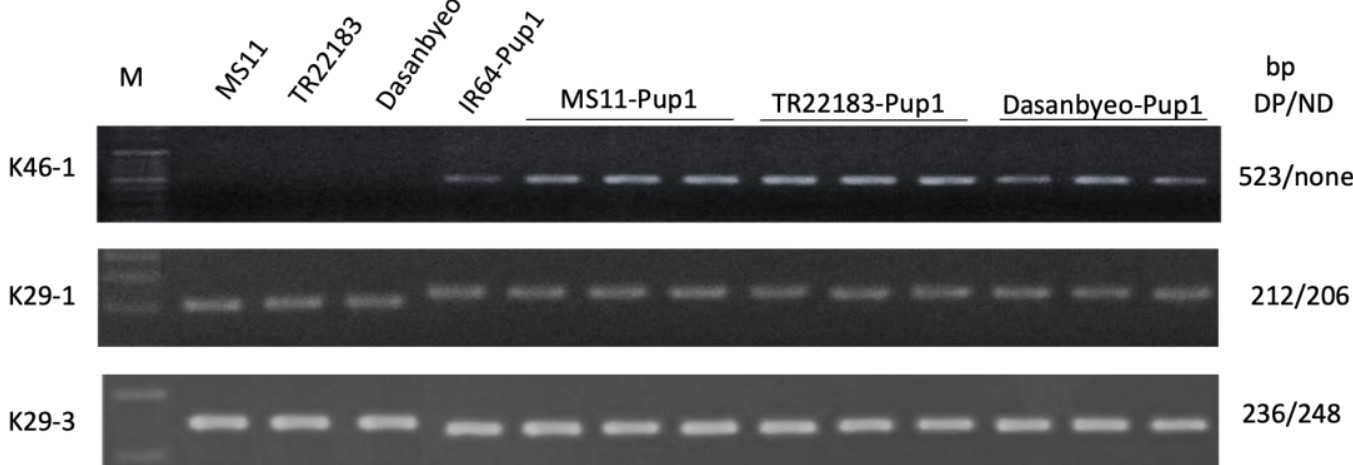

**Figure 2.** Homozygous full *Pup1* locus marker validation at $BC_2F_{12}$, pre-replicated yield trials under low P and water inputs using a core marker set consisting of K46-1, K29-1, and K29-3 gel-based markers. M—DNA size marker (1kb+); bp—base pair; DP—donor parent allele; ND—recurrent (non-donor) parent allele.

**Table 1.** RP genome recovery rates of the selected temperate *Pup1* introgression rice lines. Recovery ratio was calculated as the number of markers with RP alleles within 1 Mb window for each chromosome relative to the total number of markers. RP represents the recurrent parents. "*" denotes the number of crosses to a parent.

| Line | Pedigree | Generation | Genomic Similarity to RP |
|---|---|---|---|
| MS11 | RP | RP | N/A |
| MS11-Pup1A | (MS11*3/IR64-Pup1)-8-24-11-5-2-1-1-1-1-1-4 | $BC_2F_{12}$ | 71.68% |
| MS11-Pup1B | (MS11*3/IR64-Pup1)-17-15-8-3-5-4-1-1-1-1-4 | $BC_2F_{12}$ | 66.50% |
| TR22183 | RP | RP | N/A |
| TR22183-Pup1 | (TR22183*3/IR64-Pup1)-1-2-24-2-4-1-1-1-1-1-4 | $BC_2F_{12}$ | 99.75% |
| Dasanbyeo | RP | RP | N/A |
| Dasanbyeo-Pup1 | (Dasanbyeo*3/IR64-Pup1)-2-6-5-3-3-3-1-1-1-1-4 | $BC_2F_{12}$ | 92.33% |

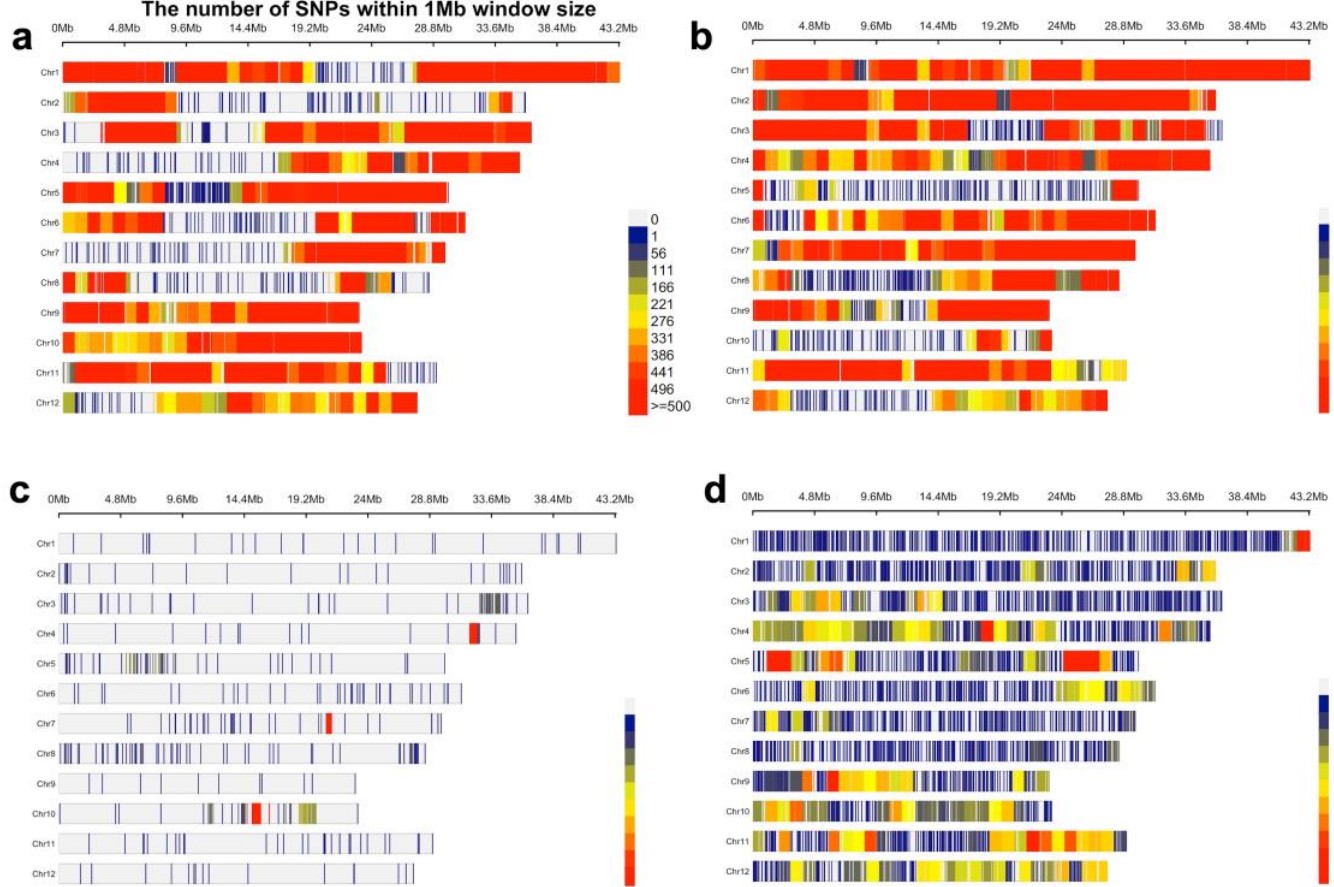

**Figure 3.** Genetic background analysis of Pup1 introgression lines, MS11-Pup1A (**a**); MS11-Pup1B (**b**); TR22183-Pup1 (**c**); Dasanbyeo-Pup1 (**d**) at $BC_2F_{12}$ using the KNU Axiom Oryza 580 K Genotyping Array containing SNPs polymorphic between RP and DP with reference to IRGSPv1.0.

The TR22183-Pup1 grown under low inputs of water and P fertilizer did not show a difference in *ph* compared to the TR22183 except at 30 DAT, where the former's *ph* was 11% greater than that of the latter. The same trend was shown on *tn*, with the TR22183-Pup1 having 20% more tillers than the TR22183 at 30 DAT. The TR22183-Pup1 had consistently more tillers than the TR22183 in P-supplied conditions. In the same treatment, the *tn* in TR22183-Pup1 (5.67) lines was 54% more than that of the TR22183 (3.67) at 54 DAT. Regardless of P treatment, SPAD readings from the TR22183 were consistently higher than that of the TR22183-Pup1 at 44 (50.97 vs. 52.80) and 54 DAT (41.53 vs. 37.77).

The Dasanbyeo-Pup1 was shorter than the Dasanbyeo under low P and water inputs at all timepoints, and it was shown to be similar in P-supplied conditions, except for at 54 DAT. The Dasanbyeo-Pup1 did not show significant difference to Dasanbyeo in terms of the *tn* in P-non-supplied conditions, whereas in P-supplied soil, it had 20% and 28% more tillers than the Dasanbyeo at 30 and 54 DAT, respectively. The SPAD values between the Dasanbyeo-Pup1 and the Dasanbyeo were only significantly different at 44 DAT in both P-supplied and P-non-supplied soils.

As an *indica* control, we evaluated a near-isogenic line (NIL) harboring full *Pup1* in the IR64 background under low P and water input. The RP IR64 showed greater *ph* than the IR64-Pup1 at 44 (41.33 cm vs. 35.00 cm) and 54 (42.33 cm vs. 37.00 cm) DAT. Under P-non-supplied conditions, there was no significant difference between the IR64-Pup1 and the IR64, whereas in P-supplied conditions, the IR64-Pup1 had consistently more tillers than the IR64 at 30 (5.00 vs. 4.33) 44, (12.00 vs. 8.67), and 54 (12.67 vs. 10.50) DAT.

**Table 2.** Vegetative stage response of Temperate-Pup1 introgression lines and IR64-Pup1 (*indica* control) grown in 2022 at Hwaseong low-phosphorus experimental field. Means with the same letters within a genetic background, treatment, and timepoint are not significantly different using Duncan's Multiple Range Test (DMRT) at $p = 0.05$. Values are means ± Standard error of the mean (SE).

| Line | Treatment | Plant Height (cm) | | | Tiller Number | | | SPAD Value | | |
|---|---|---|---|---|---|---|---|---|---|---|
| | | 30 DAT | 44 DAT | 54 DAT | 30 DAT | 44 DAT | 54 DAT | 30 DAT | 44 DAT | 54 DAT |
| MS11 | | 39.53 [a] ± 0.21 | 55.50 [a] ± 1.08 | 53.50 [b] ± 1.14 | 3.50 ± 0.43 | 4.00 [b] ± 0.76 | 5.67 [ab] ± 1.15 | 38.45 [b] ± 0.66 | 35.18 [b] ± 0.93 | 30.95 [b] ± 1.36 |
| MS11-Pup1A | | 38.00 [a] ± 0.58 | 50.67 [b] ± 2.52 | 61.67 [a] ± 1.53 | 3.33 ± 1.00 | 5.00 [ab] ± 0.58 | 5.00 [b] ± 0.67 | 36.27 [b] ± 1.95 | 31.67 [b] ± 2.99 | 34.19 [ab] ± 2.17 |
| MS11-Pup1B | | 34.83 [b] ± 1.67 | 44.67 [c] ± 2.03 | 50.33 [b] ± 1.67 | 3.00 ± 0.88 | 5.33 [a] ± 1.53 | 6.33 [a] ± 0.88 | 46.30 [a] ± 1.21 | 39.50 [a] ± 2.70 | 36.77 [a] ± 1.27 |
| TR22183 | P | 38.00 [a] ± 1.00 | 58.00 ± 5.13 | 72.33 ± 1.86 | 1.67 [b] ± 0.33 | 2.33 ± 0.33 | 3.00 ± 0.58 | 44.87 ± 0.81 | 50.97 [a] ± 0.98 | 41.53 [a] ± 0.78 |
| TR22183-Pup1 | non-supplied | 34.00 [b] ± 1.00 | 55.00 ± 1.15 | 69.33 ± 1.20 | 2.00 [a] ± 0.00 | 2.67 ± 0.33 | 3.00 ± 0.00 | 44.77 ± 1.92 | 42.80 [b] ± 2.25 | 37.77 [b] ± 1.72 |
| Dasanbyeo | | 31.67 ± 0.71 | 43.50 [a] ± 2.26 | 51.00 [a] ± 2.45 | 3.17 ± 0.17 | 4.67 ± 0.33 | 5.50 ± 0.43 | 44.20 ± 1.22 | 42.18 [b] ± 1.06 | 39.17 ± 1.46 |
| Dasanbyeo-Pup1 | | 30.00 ± 1.15 | 38.67 [b] ± 0.88 | 46.33 [b] ± 0.67 | 2.33 ± 0.67 | 4.33 ± 0.67 | 5.00 ± 0.58 | 45.43 ± 2.13 | 44.50 [a] ± 1.10 | 37.57 ± 0.93 |
| IR64 | | 34.33 ± 1.34 | 41.33 [a] ± 0.34 | 42.33 [a] ± 1.15 | 3.50 ± 0.50 | 4.00 ± 0.26 | 6.83 ± 0.75 | 43.43 [a] ± 0.55 | 41.92 ± 0.81 | 38.35 ± 1.79 |
| IR64-Pup1 | | 32.17 ± 0.67 | 35.67 [b] ± 2.67 | 37.00 [b] ± 0.67 | 3.00 ± 0.00 | 3.67 ± 0.67 | 7.00 ± 0.58 | 39.90 [b] ± 0.61 | 42.27 ± 0.67 | 39.03 ± 3.52 |
| MS11 | | 38.67 [b] ± 1.33 | 56.17 [b] ± 3.21 | 62.83 [a] ± 2.52 | 5.50 [c] ± 0.33 | 7.67 [a] ± 0.58 | 7.50 [b] ± 0.33 | 40.45 ± 1.46 | 33.57 [b] ± 1.43 | 30.57 [b] ± 2.20 |
| MS11-Pup1A | | 40.00 [a] ± 1.33 | 63.00 [a] ± 4.26 | 66.00 [a] ± 2.03 | 7.00 [a] ± 0.00 | 10.00 [b] ± 0.67 | 10.67 [a] ± 0.58 | 42.17 ± 1.73 | 34.33 [b] ± 1.65 | 31.03 [b] ± 1.48 |
| MS11-Pup1B | | 35.67 [b] ± 3.06 | 51.33 [b] ± 0.33 | 56.67 [b] ± 1.67 | 6.33 [b] ± 0.33 | 11.00 [b] ± 1.15 | 11.33 [a] ± 0.58 | 43.70 ± 2.22 | 39.57 [a] ± 1.99 | 38.17 [a] ± 1.74 |
| TR22183 | P | 38.33 ± 0.88 | 60.33 ± 0.67 | 71.67 ± 3.18 | 3.00 [a] ± 0.00 | 4.00 [a] ± 0.00 | 3.67 [b] ± 1.20 | 48.53 ± 1.41 | 46.67 [a] ± 8.16 | 45.20 [a] ± 1.42 |
| TR22183-Pup1 | supplied | 39.67 ± 1.20 | 61.67 ± 2.03 | 70.67 ± 1.45 | 4.33 [b] ± 0.33 | 5.33 [b] ± 0.33 | 5.67 [a] ± 1.20 | 49.57 ± 2.08 | 42.20 [b] ± 0.68 | 38.73 [b] ± 3.69 |
| Dasanbyeo | | 34.10 [a] ± 0.43 | 49.08 [a] ± 1.20 | 53.33 ± 0.95 | 4.17 [b] ± 0.17 | 8.17 ± 0.60 | 7.83 [b] ± 0.65 | 44.93 ± 1.13 | 40.88 [b] ± 1.33 | 34.83 ± 1.62 |
| Dasanbyeo-Pup1 | | 30.67 [b] ± 0.33 | 46.33 [b] ± 0.33 | 52.83 ± 4.11 | 5.00 [a] ± 0.00 | 8.33 ± 1.20 | 10.00 [a] ± 0.58 | 46.31 ± 0.93 | 44.39 [a] ± 0.83 | 37.50 ± 0.71 |
| IR64 | | 37.67 ± 0.52 | 47.33 [b] ± 0.98 | 47.33 ± 1.56 | 4.33 [b] ± 0.42 | 8.67 [b] ± 1.67 | 10.50 [b] ± 0.43 | 43.47 ± 1.50 | 43.85 ± 0.98 | 37.57 [a] ± 0.93 |
| IR64-Pup1 | | 36.33 ± 2.33 | 50.67 [a] ± 0.88 | 48.67 ± 1.76 | 5.00 [a] ± 0.58 | 12.00 [a] ± 0.00 | 12.67 [a] ± 0.67 | 41.97 ± 3.56 | 42.27 ± 0.73 | 34.77 [b] ± 0.19 |

To initially evaluate the performance of some introgression lines under P-supplied rainfed conditions, a preliminary experiment was conducted in 2021. Under rainfed conditions, the MS11-Pup1 introgression line was shorter than the MS11 at all time points. At 60 DAT, both the MS11-Pup1A and the MS11-Pup1B had significantly more tillers than the MS11, with the MS11-Pup1B (25.27) having twice more than the MS11 (12.73) (Table 3). In the same experiment, the TR22183-Pup1 was as tall as TR22183 and had 73% and 53% more tillers than the RP at 40 and 60 DAT, respectively. Both the MS11 and TR22183-Pup1 introgression lines showed apparent better vigor than the RP (Figure 4).

**Table 3.** Vegetative stage response of Temperate-Pup1 introgression lines under rainfed and irrigated conditions (2021). Means with the same letters within a genetic background, treatment, and timepoint are not significantly different using DMRT at $p$ = 0.05. Values are means ± SE.

| Line | Rainfed | | | | Irrigated | | | |
| --- | --- | --- | --- | --- | --- | --- | --- | --- |
| | Plant Height (cm) | | Tiller Number | | Plant Height (cm) | | Tiller Number | |
| | 40 DAT | 60 DAT | 40 DAT | 60 DAT | 40 DAT | 60 DAT | 40 DAT | 60 DAT |
| MS11 | 69.90 [a] ± 0.67 | 88.73 [a] ± 1.25 | 11.73 [b] ± 0.52 | 12.73 [c] ± 0.46 | 78.00 ± 1.22 | 99.10 [a] ± 0.62 | 14.80 [b] ± 0.58 | 12.80 ± 1.02 |
| MS11-Pup1A | 69.13 [a] ± 1.55 | 83.33 [b] ± 0.99 | 17.33 [c] ± 0.80 | 17.27 [b] ± 0.56 | 77.00 ± 0.45 | 100.00 [a] ± 1.38 | 19.60 [a] ± 1.75 | 13.40 ± 1.03 |
| MS11-Pup1B | 51.87 [b] ± 0.96 | 69.13 [c] ± 1.58 | 25.93 [a] ± 0.63 | 25.27 [a] ± 0.86 | 76.00 ± 0.71 | 88.20 [b] ± 2.85 | 18.80 [a] ± 0.86 | 14.00 ± 0.84 |
| TR22183 | 70.63 ± 1.95 | 86.67 ± 2.76 | 5.33 [b] ± 0.36 | 6.87 [b] ± 0.32 | 87.80 [a] ± 1.36 | 106.70 ± 3.28 | 9.20 ± 0.49 | 9.60 ± 0.40 |
| TR22183-Pup1 | 69.53 ± 1.22 | 84.00 ± 0.93 | 9.20 [a] ± 0.54 | 10.53 [a] ± 0.79 | 75.20 [b] ± 1.32 | 109.00 ± 1.01 | 9.40 ± 0.51 | 8.20 ± 0.58 |

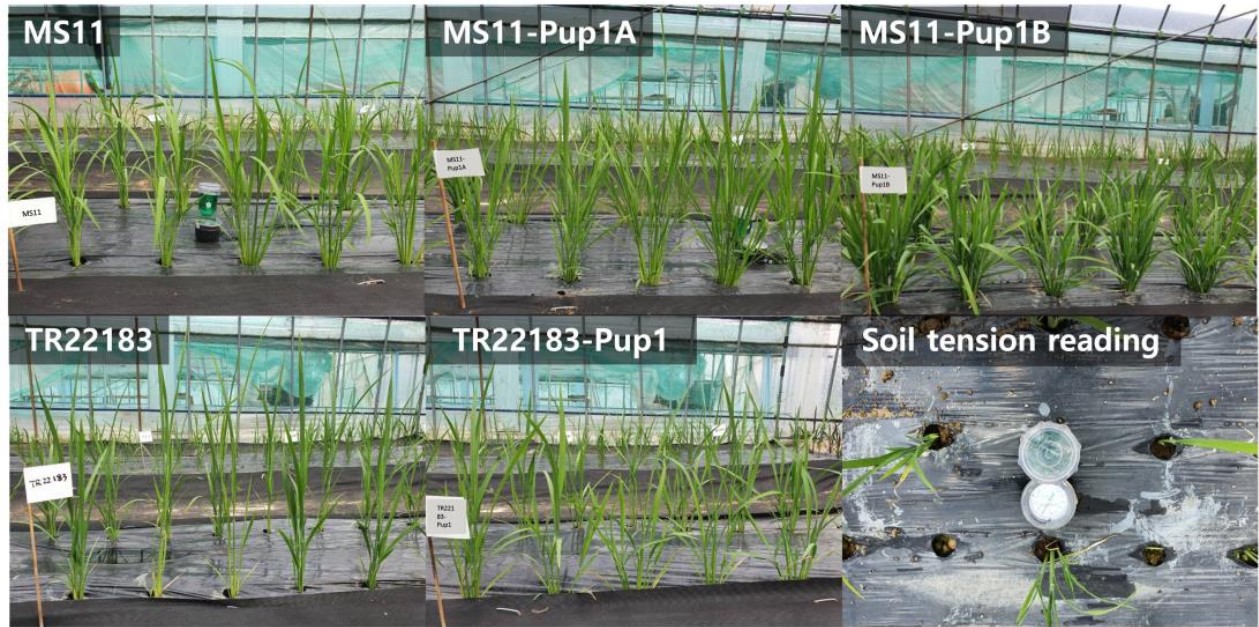

**Figure 4.** Early vigor detected in Temperate-Pup1 introgression lines under rainfed conditions (2021).

### 3.3. Yield Response to Low P and Rainfed Condition

To assess the effect of *Pup1* introgression in temperate rice in terms of yield under low inputs of P and water, important yield components were analyzed (Table 4). Plants grown in low inputs of water exhibited later heading dates, regardless of P application levels and genotypes, during the P–water regime experiment in 2022.

The MS11-Pup1A showed longer culm (77.50 cm) than both the MS11-Pup1B (66 cm) and the MS11 (70 cm) in P-non-supplied conditions, whereas culm length (*cl*) in P-supplied conditions exhibited no difference among genotypes. Both of the MS11-Pup1 introgression lines had less productive tillers in the P-non-supplied treatment. The MS11-Pup1A had less productive tillers compared to the MS11 and the MS11-Pup1B in P-supplied soil. Panicle

lengths (*pl*) in both of the MS11-Pup1 introgression lines were significantly longer than that of the MS11 at both levels of P inputs. Under P-non-supplied conditions, the panicles of both the MS11-Pup1A and MS11-Pup1B were at least 12% longer than that of the MS11. In P-supplied soil, relative to the MS11, the MS11-Pup1B and MS11-Pup1A had panicles which were longer by 25% and 12%, respectively. Fertility was not affected by the MS11 genotypes in P-non-supplied conditions. On the other hand, MS11-Pup1A's fertility rate (90.53%) was better than that of MS11-Pup1B (80.91%) and MS11 (85.61%).

**Table 4.** Reproductive stage response of temperate-Pup1 introgression lines and control varieties grown under rainfed and irrigated conditions (2022). Means with the same letters within a genetic background and treatment not significantly different using DMRT at *p* = 0.05, except for days to heading. Values are means ± SE.

| Line | Treatment | Days to Heading (DAT) | Culm Length (cm) | Productive Tiller Number | Panicle Length (mm) | Hundred Grain Weight (g) | Fertility (%) |
|---|---|---|---|---|---|---|---|
| MS11 | | 78 | 70.80 [b] ± 2.27 | 7.00 [a] ± 0.63 | 223.00 [b] ± 7.00 | 2.86 ± 0.05 | 94.04 ± 2.18 |
| MS11-Pup1A | | 76 | 77.50 [a] ± 0.92 | 4.60 [b] ± 0.24 | 251.12 [a] ± 5.10 | 2.42 ± 0.13 | 94.01 ± 2.21 |
| MS11-Pup1B | P non-supplied | 82 | 66.00 [c] ± 0.45 | 5.40 [b] ± 0.24 | 251.61 [a] ± 5.20 | 2.62 ± 0.02 | 94.60 ± 3.46 |
| TR22183 | | 51 | 51.50 [b] ± 0.39 | 3.20 ± 0.20 | 254.21 [b] ± 4.20 | 2.82 ± 0.11 | 92.42 [b] ± 4.59 |
| TR22183-Pup1 | | 58 | 56.00 [a] ± 0.84 | 3.40 ± 0.51 | 267.10 [a] ± 3.30 | 2.91 ± 0.04 | 100.00 [a] ± 0.00 |
| Dasanbyeo | | 84 | 56.50 ± 1.20 | 4.20 ± 0.20 | 260.02 ± 3.51 | 2.70 ± 0.09 | 90.33 ± 2.94 |
| Dasanbyeo-Pup1 | | 91 | 53.20 ± 0.20 | 3.80 ± 0.37 | 261.61 ± 6.72 | 2.79 ± 0.08 | 95.43 ± 2.16 |
| MS11 | | 68 | 68.40 ± 2.07 | 9.20 [b] ± 0.86 | 209.02 [c] ± 4.60 | 2.57 ± 0.12 | 85.61 [ab] ± 2.75 |
| MS11-Pup1A | | 72 | 73.30 ± 1.84 | 7.20 [a] ± 0.20 | 234.05 [b] ± 9.01 | 2.49 ± 0.04 | 90.53 [a] ± 4.17 |
| MS11-Pup1B | P supplied | 78 | 70.40 ± 1.21 | 9.20 [b] ± 0.58 | 261.02 [a] ± 8.05 | 2.66 ± 0.06 | 80.91 [b] ± 4.60 |
| TR22183 | | 56 | 59.40 [b] ± 1.71 | 4.80 ± 0.37 | 243.00 [b] ± 13.32 | 2.89 ± 0.04 | 99.17 [a] ± 0.83 |
| TR22183-Pup1 | | 65 | 64.90 [a] ± 1.25 | 5.20 ± 0.66 | 266.00 [a] ± 8.31 | 2.81 ± 0.05 | 95.19 [b] ± 1.54 |
| Dasanbyeo | | 82 | 69.30 [a] ± 0.86 | 7.00 ± 0.71 | 251.02 ± 5.88 | 2.63 ± 0.07 | 96.86 ± 1.59 |
| Dasanbyeo-Pup1 | | 82 | 61.50 [b] ± 0.32 | 6.20 ± 0.20 | 245.10 ± 3.87 | 2.83 ± 0.05 | 94.01 ± 1.84 |

TR22183-Pup1's culm was longer than TR22183's, regardless of the P–water levels. Under P-non-supplied condition, the panicles of the TR22183-Pup1 were significantly longer (267.10 mm) than that of the TR22183 (254.21 mm), whereas in P-supplied conditions, no significant difference was observed. The TR22183-Pup1 had higher fertility than the TR22183 (100% vs. 92%), whereas the opposite was observed in P-supplied conditions with TR22183-Pup1 and TR22183, which had fertility rates of 95% and 99%, respectively.

Under both levels of P applications, yield components between the Dasanbyeo-Pup1 and Dasanbyeo did not differ significantly, except for culm length (P-supplied condition), with the Dasanbyeo having approximately 13% longer culms than the Dasanbyeo-Pup1.

Grain yield per plant (*gypp*) obtained from different levels of P inputs under rainfed conditions varied depending on the genetic background (Figure 5). Under P-non-supplied conditions, only the TR22183-Pup1 showed a significant yield advantage relative to the RP. The TR22183-Pup1 had 20% more *gypp* than the TR22183. Under P-supplied conditions, the MS11-Pup1B and TR22183 had significant yield advantages of 24% and 26%, respectively, relative to that of the RPs. On the other hand, the *gypp* of the Dasanbyeo-Pup1 did not differ significantly from the Dasanbyeo at both levels of P applications. Dasanbyeo is genetically close to *indica*, so we tested them with the *indica* control, IR64-Pup1 (Table S1). The IR64-Pup1 performed lower than the IR64 in terms of *gypp*, at 21.18 g (IR64-Pup1) and 31.42 g (IR64), respectively, under P-non-supplied conditions. It was caused by a lower tiller number in the IR64-Pup1. On the other hand, yield components between the IR64-Pup1 and IR64 did not differ significantly in P-supplied conditions.

To test the effect of *Pup1* introgression into temperate rice grown under rainfed versus irrigated conditions in P-supplied conditions, we analyzed the yield components obtained from an experiment conducted in 2021 using the MS11- and TR22183-Pup1 introgression lines (Table 5). The MS11-Pup1A and MS11-Pup1B panicles were longer than that of

the MS11 by 25 mm and 21 mm, respectively. The TR22813-Pup1, on the other hand, did not exhibit longer panicles compared to the RP, regardless of water input. *Pup1* introgression lines in both of the MS11 and TR22183 backgrounds exhibited significant fertility advantages relative to the RPs under rainfed conditions. The fertility rates of the MS11-Pup1A and MS11-Pup1B were 3.5 and 2.9 times better, respectively, relative to that of the MS11. TR22183-Pup1's fertility rate (66.94%) was more than 3-folds that of TR22183's (20.04%). Ultimately, the *gypp* in the MS11-Pup1A (27.31 gm) and MS11-Pup1B (26.56 gm) had values which were more than 2-folds that of the MS11. In case of the TR22183-Pup1, it had a *gypp* of 25.71 g, which was 42% higher than that of the TR22183.

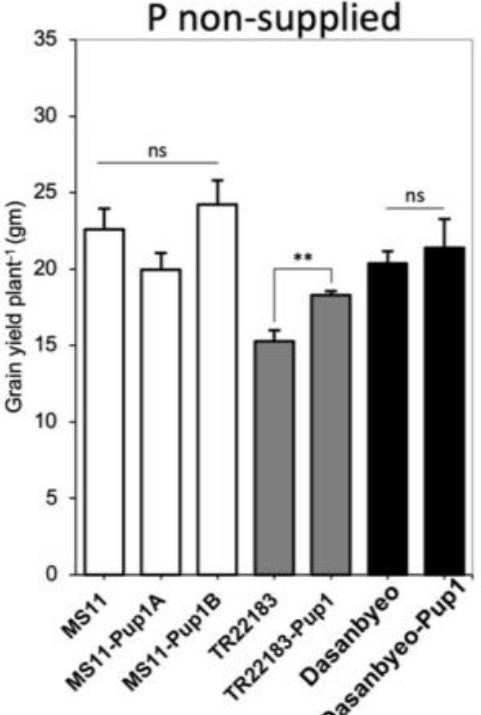 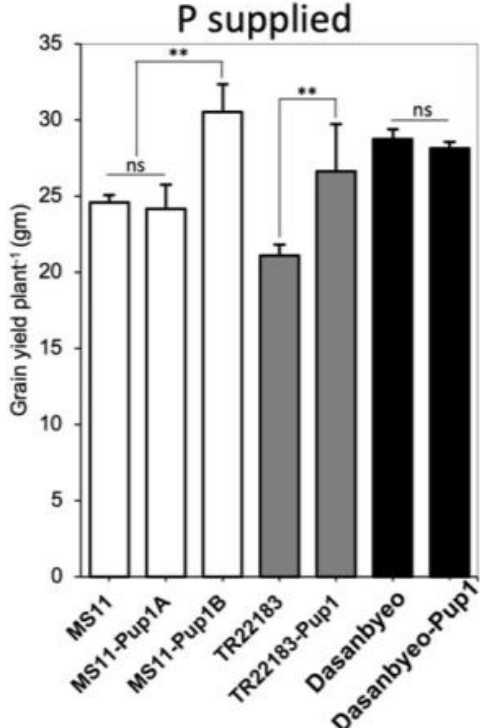

**Figure 5.** Yield response of temperate *Pup1* introgression lines grown in P-non-supplied and P-supplied soils under rainfed conditions in 2022. ** indicates statistical significance using DMRT, whereas ns denotes insignificance within a genetic background at *p* = 0.05.

**Table 5.** Reproductive stage response of temperate *Pup1* introgression lines grown under rainfed and irrigated conditions (2021). Means with the same letters within a genetic background and treatment are not significantly different using DMRT at *p* = 0.05. Values are means ± SE.

| Line | Treatment | Culm Length (cm) | Productive Tiller Number | Panicle Length (mm) | Hundred Grain Weight (g) | Fertility (%) | Grain Yield per Plant (g) |
|---|---|---|---|---|---|---|---|
| MS11 | | 47.43 [c] ± 0.60 | 14.60 ± 0.51 | 183.20 [b] ± 4.72 | 1.70 ± 0.16 | 20.06 [c] ± 2.44 | 13.29 [b] ± 1.20 |
| MS11-Pup1A | | 64.14 [a] ± 1.45 | 12.60 ± 0.75 | 208.20 [a] ± 4.44 | 1.75 ± 0.82 | 69.98 [a] ± 1.75 | 27.31 [a] ± 1.29 |
| MS11-Pup1B | Rainfed | 54.70 [b] ± 1.20 | 15.20 ± 1.02 | 204.40 [a] ± 4.12 | 1.66 ± 0.77 | 58.72 [b] ± 3.64 | 26.56 [a] ± 2.20 |
| TR22183 | | 43.44 ± 4.93 | 18.10 ± 2.20 | 236.00 ± 9.92 | 2.46 ± 0.24 | 20.04 [b] ± 5.21 | 18.10 [b] ± 3.19 |
| TR22183-Pup1 | | 52.76 ± 1.49 | 14.50 ± 1.57 | 235.20 ± 13.03 | 2.64 ± 0.99 | 66.94 [a] ± 2.74 | 25.71 [a] ± 0.87 |
| MS11 | | 79.76 ± 2.36 | 15.80 [b] ± 1.57 | 266.40 [a] ± 7.22 | 2.20 [b] ± 1.19 | 87.56 [a] ± 2.46 | 28.42 [b] ± 1.65 |
| MS11-Pup1A | | 79.76 ± 2.36 | 14.50 [b] ± 0.58 | 222.00 [b] ± 2.83 | 2.18 [b] ± 0.80 | 78.64 [b] ± 3.45 | 29.72 [b] ± 1.87 |
| MS11-Pup1B | Irrigated | 77.56 ± 2.27 | 20.60 [a] ± 1.17 | 259.20 [a] ± 8.33 | 2.78 [a] ± 1.44 | 88.58 [a] ± 1.50 | 52.02 [a] ± 2.23 |
| TR22183 | | 68.52 [b] ± 1.87 | 24.40 [a] ± 0.68 | 265.60 ± 9.50 | 2.84 ± 0.86 | 92.74 ± 1.97 | 45.64 ± 2.08 |
| TR22183-Pup1 | | 76.24 [a] ± 1.72 | 21.20 [b] ± 0.58 | 281.20 ± 2.94 | 2.83 ± 0.51 | 92.62 ± 0.97 | 42.32 ± 2.90 |

Plant shape, panicle architecture, and grain shape did not virtually differ between the *Pup1* introgression lines and the RPs, except for the MS11-Pup1 introgression lines (Figure 6). Both the MS11-Pup1A and MS11-Pup1B had longer panicles and grains compared to that of the MS11.

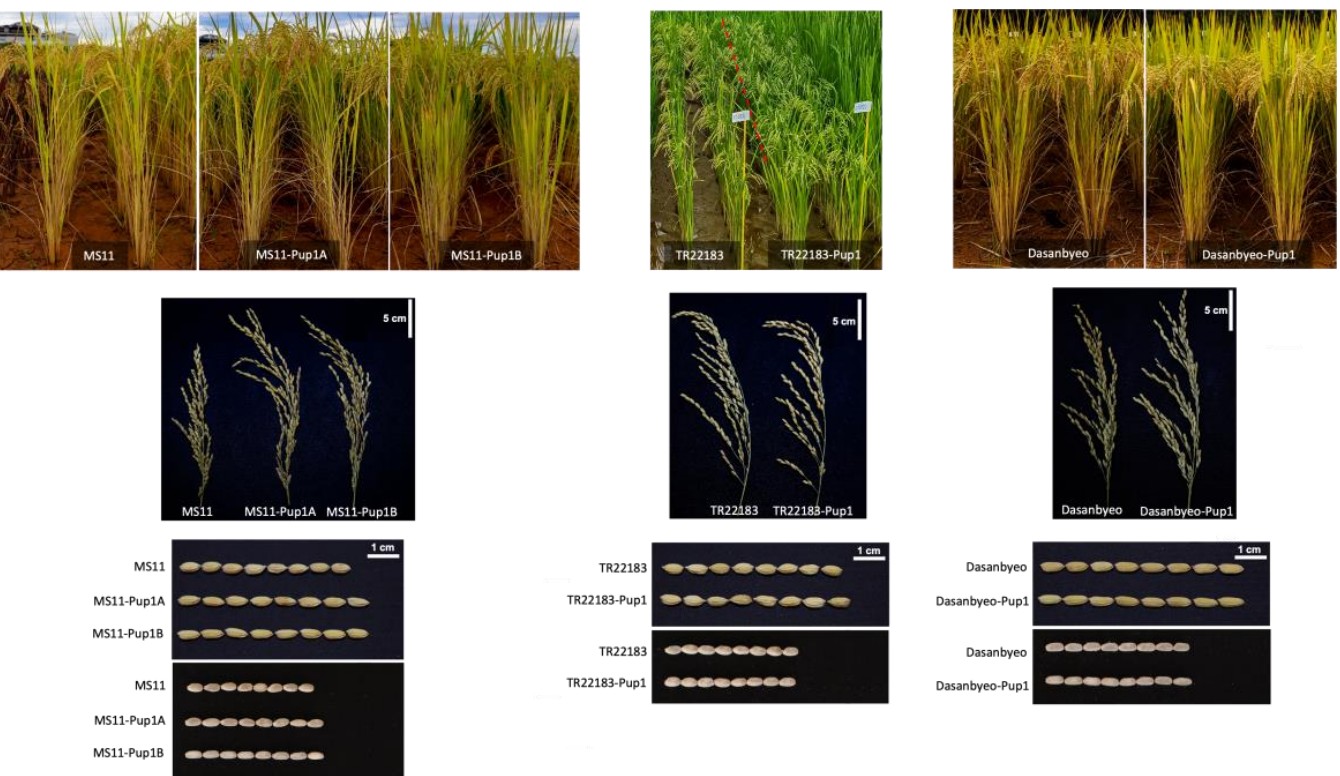

**Figure 6.** Plant shape, panicle, and grain characteristics of the temperate *Pup1* introgression lines grown in low P and water input (Hwaseong, 2022).

## 4. Discussion

Climate change and the increasing cost of agricultural inputs, such as fertilizers, are two of the most serious constraints in rice production. Genetic improvement through molecular marker-assisted breeding is one of the most sustainable and rapid approaches in developing rice that are adaptable to low agricultural inputs and a shifting climate. Recent advances in molecular plant breeding have paved the way for the development of high yielding varieties (HYVs). Modern agriculture is based on these HYVs which are bred for high yield, but also require high levels of agricultural inputs, such as irrigation, fertilizer, and pesticides. However, high-input systems are becoming less feasible in the face of natural resource decline and population growth [24]. In this study, we investigated the efficacy of *Pup1*, a P deficiency tolerance-enhancing QTL, by developing temperate rice lines with full *Pup1* introgression and evaluating those lines under low P and water inputs in a temperate climate.

The effect of P deficiency is more pronounced in rainfed rice farming systems, where the lack of water movement renders inorganic P extremely immobile in soil. Furthermore, in areas intensively cultivated for HYVs, acidic soils are predominant where loosely bound, plant-available phosphates slowly convert into highly crystalline iron, (Fe)-P, and aluminum, (Al)-P, through ligand exchange [25–27]. Here, we evaluated the response of the genotypes to two levels of P applications under rainfed and slightly acidic soil conditions (Figure S1 and Table S2).

We introgressed the full *Pup1* QTL into three temperate rice varieties—MS11, Dasanbyeo, and TR22183—using a core set of foreground markers and tagging the three most important genes in the *Pup1* region—*OsPupK46-2* or *OsPSTOL1*, *OsPupK20-2*, and *OsPupK29-*

*1— c*oupled with high density SNP-based background genotyping. *Indica* varieties with full and partial *Pup1* introgression grown under tropical climates have historically shown enhanced phosphorus uptake and yield compared to rice breeding lines not harboring the QTL [28].

*Pup1* introgression lines exhibited various vegetative response patterns to low inputs of P and water, depending on the genetic background. *Pup1* introgression lines did not show a clear advantage in terms of early vegetative growth. However, in P-supplied conditions and at maximum tillering stage, the *Pup1* lines were better than their RPs. In the experiment conducted in 2022, the available P content of the soil in the P-supplied condition was 1.9 times that of P-non-supplied soil (Table S2). Therefore, the phenotypic responses of the lines under different input levels of P might be related to the increase in P uptake when additional P fertilizer was supplied. In a previous study, *Pup1* introgression lines in the IR64 background—IR64-Pup1 and IR64-Pup1+Sub1—exhibited more than a two-fold change in shoot P content in the P-supplied content versus the P-non-supplied soil at 48 DAT compared to that of the IR64, [13] which suggests *Pup1*'s benefit when P fertilizer is supplemented.

We evaluated the yield performance of the *Pup1* introgression lines under rainfed conditions in P-supplied condition. *Pup1* introgression lines in the MS11 and TR22183 background exhibited yield advantages over the RPs that were mostly attributable to the improved fertility (Table 5). Presumably, the *Pup1* QTL may have conferred improved yield performance under rainfed conditions through the improvement of root architecture at early growth stages that led to improved fertility. *OsPSTOL1*, the main causal gene in the *Pup1* region, encodes a protein kinase that improves root architecture in rice to scavenge more inorganic P (Pi) in the topsoil region during the early growth stage. Interestingly, *OsPSTOL1's* putative downstream genes co-collocate with QTLs associated with root and drought tolerance, which is associated with the altered expression of genes related to yield under drought [23]. This claim, however, needs to be validated by investigating root phenotypes in the early stages of growth.

The TR22183-Pup1 and MS11-Pup1B consistently had better yield at low P and water inputs across the two cropping seasons (2021 and 2022). In P deficient soils, the *Pup1* QTL increased yield in rice by enhancing root architecture, tillering capacity, and grain filling [24]. In this study, rather than having tolerance to low P, *Pup1* seemed to enhance P uptake when P fertilizer was applied. Among the yield components evaluated, the enhancement in the panicle length was a clear index for improved yield under low P application. The panicles of the TR22183- and MS11-Pup1 introgression lines were longer than the RPs, which translated to yield advantages over the TR22183 and MS11 (Table 4, Figure 5). In terms of grain yield response to additional P fertilizer, MS11-Pup1A, MS11-Pup1B and TR22183-Pup1 had more advantages than the RPs by a factor of 2.3 times, 2.9 times and 1.2 times, respectively (Table 6). In a previous study, tropical region-specific panicle length and yield component QTLs were detected in RILs derived from a cross between a temperate *japonica* and a Tongil-type rice. The QTLs were not detected when the same set of RILs were grown in a temperate region under low inputs of water and P fertilizer [29]. This suggests that panicle length might be an important index for the genetic improvement of rice adaptability to climate change shifts and low input systems.

The Dasanbyeo-Pup1 did not have advantage over the Dasanbyeo, both in terms of vegetative growth and yield, regardless of the P application. The Dasanbyeo was derived from an intercross between IR8, Taichung Native 1, and Yukara. It is also closely related to *indica*-type varieties [15]. *Indica* rice is adaptable to the tropics, whereas *japonicas* are mainly grown in temperate regions [30]. In this study, an *indica* variety, IR64-Pup1, which originally performed well in the tropical region, was not better than the IR64 in terms of vegetative growth and yield. It performed even worse than the IR64 in P-non-supplied soil (Supplementary Table S4). Since both the Dasanbyeo and IR64 have later days to heading and maturity dates in temperate climate due to photosensitivity, the expression of *Pup1* might have been masked due to the longer duration of growth in *indica* varieties, and

therefore longer exposure to low available phosphorus in the soil. The unexpected response of the Dasanbyeo-Pup1 and IR64-Pup1 necessitates further studies on the expression levels of the stress tolerance enhancing genes, which have been previously known to work in tropical conditions with a different time frame of growth. This suggests a need to develop rice varieties with less photosensitivity and excellent early vigor for the enhanced utilization of water and inorganic fertilizers.

**Table 6.** Relative response of the temperate-Pup1 breeding lines to P fertilizer application under rainfed conditions. Values were obtained using the formula (Response to P-supplied condition–response to P-non-supplied condition)/(Response to P-non-supplied condition).

| Line | Days to Heading (DAT) | Culm Length (cm) | Productive Tiller Number | Panicle Length (cm) | Hundred Grain Weight (g) | Fertility (%) | Grain Yield per Plant (g) |
|---|---|---|---|---|---|---|---|
| MS11 | −0.13 | −0.03 | 0.31 | −0.06 | −0.10 | −0.09 | 0.09 |
| MS11-Pup1A | −0.05 | −0.05 | 0.57 | −0.07 | 0.03 | −0.04 | 0.21 |
| MS11-Pup1B | −0.05 | 0.07 | 0.70 | 0.04 | 0.02 | −0.14 | 0.26 |
| TR22183 | 0.10 | 0.15 | 0.50 | 0.05 | 0.02 | 0.07 | 0.38 |
| TR22183-Pup1 | 0.12 | 0.16 | 0.53 | −0.09 | −0.03 | −0.05 | 0.46 |
| Dasanbyeo | −0.02 | 0.23 | 0.67 | −0.03 | −0.03 | 0.07 | 0.41 |
| Dasanbyeo-Pup1 | −0.10 | 0.16 | 0.63 | −0.06 | 0.01 | −0.01 | 0.32 |

## 5. Conclusions

We introgressed the P deficiency tolerance enhancing *Pup1* QTL into three temperate varieties—MS11, TR22183, and Dasanbyeo—to evaluate whether the QTL works in temperate rice grown under a temperate climate. To conclude, *Pup1* enhanced P uptake in temperate rice with a *japonica* background, whereas introgression into *indica* rice (i.e., Dasanbyeo and IR64) did not show significant effects. The genetic background-dependent differences in the level of efficacy of the *Pup1* QTL observed in this study necessitates further studies on the differences in gene networks among rice subspecies which might be involved in the expression of genes conferring abiotic stress tolerance in rice. Our study established a preliminary basis for how breeders can precisely choose genetic backgrounds for *Pup1* breeding programs in specific ecotypes.

**Supplementary Materials:** The following supporting information can be downloaded at: https://www.mdpi.com/article/10.3390/agriculture12122056/s1, Figure S1: Soil tension readings in the P–water experimental field in Hwaseong (2022).; Table S1: Reproductive stage response of Dasan-Pup1, Dasanbyeo, and *indica* controls IR64-Pup1 and IR64 grown under rainfed conditions in two levels of P fertilization (2022).; Table S2: Chemical and physical properties of the soil used in the experiment in Hwaseong in 2022.

**Author Contributions:** Conceptualization, J.H.C., S.-W.K. and I.P.N.; methodology, J.H.C., N.-H.S., I.P.N. and J.-H.H.; investigation, J.-H.H., N.-H.S., O.N.L. and I.P.N.; data curation, J.-H.H., N.-H.S. and I.P.N.; formal analysis, I.P.N., J.-H.H. and O.N.L.; funding acquisition, J.H.C.; project administration, J.H.C.; resources, J.H.C. and S.-W.K.; software, I.P.N. and J.-H.H.; supervision, J.H.C.; validation, I.P.N. and J.H.C.; visualization, I.P.N., J.-H.H. and O.N.L.; writing—original draft, I.P.N.; writing—review and editing, J.H.C., I.P.N., N.-H.S., J.-H.H., S.-W.K. and I.-R.C. All authors have read and agreed to the published version of the manuscript.

**Funding:** This work was supported under the framework of the international cooperation program managed by the National Research Foundation of Korea (NRF-2021K1A3A1A61002988). This work was supported by the Temperate Rice Research Consortium (TRRC) of the International Rice Research Institute (IRRI).

**Institutional Review Board Statement:** Not applicable.

**Data Availability Statement:** Not applicable.

**Conflicts of Interest:** The authors declare no conflict of interest.

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
