# Peer review of "Assessing the Effect of a Major Quantitative Locus for Phosphorus Uptake (Pup1) in Rice (O. sativa L.) Grown under a Temperate Region"

_agriculture, doi:10.3390/agriculture12122056_

Round 1
Reviewer 1 Report
This manuscript presents introgression of a high phosphorus uptake allele of Pup1 from indica rice variety IR64-Pup1 into three temperate rice varieties and evaluation of the introgression lines. Results indicate that the Pup1 allele could enhance phosphorus uptake in temperate rice and the efficacy may vary depending on the genetic background. Improvements are required before it is accepted for publication in Agriculture.
Comments and suggestions:
Lines 119-121: “Ten F1 plants from each cross were subjected to foreground genotyping using the K29-1 codominant Pup1 marker instead of the K46-1 marker, because the K46-1 marker is a dominant type, making it challenging to select plants containing heterozygous alleles”. Not clear. F1 plants naturally have the heterozygous alleles.
Lines 123-124: “were foreground-genotyped using K46-1, K29-3, and K29-1”. Different from the description in Lines 119-121. Why use K46-1? There are also other explanations on choosing markers in Lines 236-238. All these should be clearly presented in one place in M&M.
Lines 128-146: Are the rice lines from MS11 subjected to background identification? Figure 1 shows no difference among the three populations. Also, not appropriate to describe “BC2F1 plants in TR22183 and Dasanbyeo backgrounds were background genotyped using the 6K Illumina genotyping platform” and “In case of MS11-Pup1, early generation background genotyping was not conducted. Instead, pedigree selection was done based on the morphological similarity to the RP” in Results (Lines 240-242, 247-249). All these should be clearly presented in one place in M&M. There are others of this kind, please check carefully and revise.
The numbers of SNPs/polymorphic SNPs are confusing. In the text, it is said “Of the total 5,274 SNP markers, 2,055 were polymorphic between TR22183 and IR64-Pup1, whereas 146 markers were polymorphic between Dasanbyeo and IR64-Pup1”. In Figure 3, there are many “1 Mb size window” with red color that has SNP numbers ranging from 386/441 to >500. Please note that a recovery rate is determined based on polymorphic SNPs.
Discussion: could be more concise by not talking much on those already presented in M&M and Results.
Author Response
Please refer below for the response to the reviewer:
Reviewer 1
Comment 1: “Lines 119-121: “Ten F1 plants from each cross were subjected to foreground genotyping using the K29-1 codominant Pup1 marker instead of the K46-1 marker, because the K46-1 marker is a dominant type, making it challenging to select plants containing heterozygous alleles”. Not clear. F1 plants naturally have the heterozygous alleles.”
Our response: We agree with the comment and have rephrased the sentence more precisely. Briefly, K29-1 was the marker of choice because it is a codominant marker (as opposed to K46-1, which is a dominant marker).
Comment 2: “Lines 123-124: “were foreground-genotyped using K46-1, K29-3, and K29-1”. Different from the description in Lines 119-121. Why use K46-1? There are also other explanations on choosing markers in Lines 236-238. All these should be clearly presented in one place in M&M.”
Our response: We agree with the comment and have incorporated changes in parts containing marker information. We have stated the reasons why these core marker set were chosen as foreground markers. Briefly, these markers are tagging the three important genes in the Pup1 locus namely, OsPSTOL1 (K46-1), OsPupK20-2 (K29-3), and OsPupK29-1 (K29-1). All these markers were employed to have introgression lines with “full” Pup1 introgression.
Comment 3: “Lines 128-146: Are the rice lines from MS11 subjected to background identification? Figure 1 shows no difference among the three populations. Also, not appropriate to describe “BC2F1 plants in TR22183 and Dasanbyeo backgrounds were background genotyped using the 6K Illumina genotyping platform” and “In case of MS11-Pup1, early generation background genotyping was not conducted. Instead, pedigree selection was done based on the morphological similarity to the RP” in Results (Lines 240-242, 247-249). All these should be clearly presented in one place in M&M. There are others of this kind, please check carefully and revise.”
Our response: We agree and have made changes in parts with more clarifications needed. We have revised figure 1. We did early generation (BC2F1) background genotyping (6K Illumina platform) ONLY for TR22183 and Dasanbyeo Pup1 introgression lines. Axium Oryza 580K was used to assess the background genome recovery rate pre yield trial in BC2F12 generation for TR22183, Dasanbyeo, and MS11.
Comment 4: “The numbers of SNPs/polymorphic SNPs are confusing. In the text, it is said “Of the total 5,274 SNP markers, 2,055 were polymorphic between TR22183 and IR64-Pup1, whereas 146 markers were polymorphic between Dasanbyeo and IR64-Pup1”. In Figure 3, there are many “1 Mb size window” with red color that has SNP numbers ranging from 386/441 to >500. Please note that a recovery rate is determined based on polymorphic SNPs.”
Our response: We agree with the suggestion to clarify more on the number of SNPs used in the background genotyping step. We have added supplementary data on the recurrent parent genome recovery rates at the early generation (BC2F1) - Illumina 6K platform data and information on the total number of polymorphic markers between Pup1 donors and recipient (BC2F12) - Axium 580K platform – Please refer to the revised supplementary (Table S3). We have added important sentences stating that only the polymorphic markers between the donor and recipient parents were used to estimate the background genome recovery rates.
Comment 5: “Discussion: could be more concise by not talking much on those already presented in M&M and Results.”
Our response: Thank you for this important comment. We have restructured the discussion part based on this suggestion.
In addition, we have re-checked for spelling and grammatical errors and have added more precise information in the materials and methods as well as in the results part.

Reviewer 2 Report
The manuscript “Assessing the effect of a major quantitative locus for phosphorus uptake (Pup1) in rice (O. sativa L.) grown under temperate region” deals with the genomic analysis of phosphorus uptake (Pup1) in rice which is grown under low temperature/ temperate conditions. Overall the manuscript is well written, planned, and executed. I have enjoyed reading this research manuscript, and indeed this manuscript will help the researchers to develop cultivar of rice which can be grown in the non-conventional area or temperate area. However, some issues need to be addressed before final decision is made.
Comment
· LN 18-19: Please rewrite the line.
· LN 22: Mention the rice cultivar is Indica, Japonica or Javanica ?
· LN 51: Please provide reference here.
· Ln 64-67: Please rewrite the line.
· LN 88-93: Please rewrite the objective of the manuscript more clearly.
· Section 2.2 is well written.
· LN 224: Write “Statistical analysis of phenotypic data” instead of “Analysis of phenotypic data”
· LN 471-475: Please rewrite the line
· I found the reference cited in the text are old enough. Kindly update some of the references with recent one. For example, reference numbers 10, 16, 19, 26, 29.
· Discussion: Current discussion is fine, but I suggest the authors split the discussion in different parts as per the results sequence and then describe each part separately.
· The conclusion section seems to be written in a hurry. Kindly revisit the conclusion section thoroughly and also mention the future thrust of the present study.
Author Response
Reviewer 2
Comment 1: “LN 18-19: Please rewrite the line.”
Our response: We agree that this line should be written with more precision and improved sentence structure. We have incorporated changes in this particular line.
Comment 2: “LN 22: Mention the rice cultivar is Indica, Japonica or Javanica”
Our response: We did not do any changes in this line as the three recipient varieties are all “temperate varieties” although Dasanbyeo is genetically close to indica. MS11 and TR22183 belong to temperate japonica.
Comment 3: “LN 51: Please provide reference here.”
Our response: Thank you for noticing. We have added appropriate reference and citation on this part.
Comment 4: “Ln 64-67: Please rewrite the line.”
Our response: We agree that this line should be written with more precision and improved sentence structure. We have incorporated changes in this particular line.
Comment 5: “LN 88-93: Please rewrite the objective of the manuscript more clearly.”
Our response: We agree that the objective should be more emphasized in these lines. We have incorporated changes in this part highlighting the research questions we want to address through this manuscript.
Comment 6: LN 224: Write “Statistical analysis of phenotypic data” instead of “Analysis of phenotypic data.”
Our response: Thank you for the suggestion. We have revised the title of this part based on your suggestion.
Comment 7: “LN 471-475: Please rewrite the line.”
Our response: We agree that this line should be written with more precision and improved sentence structure. We have incorporated changes in this particular line.
Comment 8: “I found the reference cited in the text are old enough. Kindly update some of the references with recent one. For example, reference numbers 10, 16, 19, 26, 29.”
Our response: Thank you for the suggestion. The references used in this manuscript was carefully selected in terms of relevance. We retained the original references due to its high relevance to our study and irreplaceability.
Comment 9: “Discussion: Current discussion is fine, but I suggest the authors split the discussion in different parts as per the results sequence and then describe each part separately.”
Our response: We agree with the suggestion and have incorporated changes in the discussion part based on this suggestion. We have re-arranged the discussion part and divided the parts based on the results sequence.
Comment 10: “The conclusion section seems to be written in a hurry. Kindly revisit the conclusion section thoroughly and also mention the future thrust of the present study.”
Our response: We agree with this suggestion and have re-written the conclusion part with more emphasis on the relevance of our results to actual breeding situations.
In addition, we have re-checked for spelling and grammatical errors and have added more precise information in the materials and methods as well as in the results part.
